# Experimental Study on the Effect of an Organic Matrix on Improving the Strength of Tailings Strengthened by MICP

**DOI:** 10.3390/ma16155337

**Published:** 2023-07-29

**Authors:** Lin Hu, Huaimiao Zheng, Lingling Wu, Zhijun Zhang, Qing Yu, Yakun Tian, Guicheng He

**Affiliations:** 1School of Resource & Environment and Safety Engineering, University of South China, Hengyang 421001, China; 343737982229@usc.edu.cn (L.H.); 2004001109@usc.edu.cn (H.Z.); 130000148665@usc.edu.cn (Z.Z.); 2009000547@usc.edu.cn (Q.Y.); 2017000012@usc.edu.cn (Y.T.); 2005000509@usc.edu.cn (G.H.); 2Hunan Province Engineering Technology Research Center for Disaster Prediction and Control on Mining Geotechnical Engineering, Hengyang 421001, China

**Keywords:** microbial biomineralization, tailings, organic matrix, calcium carbonate

## Abstract

In order to improve the effect of microbial-induced calcium carbonate precipitation (MICP) in tailings reinforcement, sodium citrate, an organic matrix with good water solubility, was selected as the crystal form adjustment template for inducing calcium carbonate crystallization, and the reinforcements of tailings by MICP were conducted in several experiments. The effects of sodium citrate on the yield, crystal form, crystal appearance, and distribution of calcium carbonate were analyzed by MICP solution test; thus, the related results were obtained. These showed that the addition of a proper amount of organic matrix sodium citrate could result in an increment in the yield of calcium carbonate. The growth rate of calcium carbonate reached 22.6% under the optimum amount of sodium citrate, and the crystals of calcium carbonate were diverse and closely arranged. Based on this, the MICP reinforcement test of tailings was carried out under the action of the optimum amount of sodium citrate. The microscopic analysis using CT and other means showed that the calcium carbonate is distributed more uniformly in tailings, and the porosity of samples is significantly reduced by layered scanning analysis. The results of triaxial shear tests showed that adding organic matrix sodium citrate effectively increased the cohesion, internal friction angle, and peak stress of the reinforced tailings. It aims to provide a novel idea, a creative approach, and a method to enhance the reinforcement effect of tailings and green solidification technology in the mining environment.

## 1. Introduction

Tailings refer to the remnants left after ore extraction, which are widely concerning due to their harmful substances and potential environmental risks. There are heavy metals within tailings; some can potentially permeate the soil and groundwater via natural weathering or water erosion, posing threats to biological systems and human health [1]. Additionally, tailings can disrupt the balance of ecosystems. Organic and other toxic substances within tailings can alter soil and water quality, leading to certain risks to aquatic animals and plants and the destruction of entire ecosystems. Furthermore, the storage and treatment of tailings require significant amounts of land and water resources; severe disturbances might be caused [2]. In order to minimize the detrimental impact of tailings on the environment, there is a great necessity to reduce or remove harmful substances from tailings [3,4], meanwhile ensuring by containment that they are safe and stable [5], reducing their impacts on both the ecosystem and human health. Therefore, it is of utmost importance that we urgently develop new technical approaches that make tailings reinforcement easy and efficient while remaining green and pollution-free, effectively reducing the environmental pollution risk associated with tailings.

In recent years, there has been a surge of interest in microbial-induced calcite precipitation (MICP), considered a highly effective and sustainable reinforcement technology; it has been evidenced by numerous domestic and international studies [6]. These studies have demonstrated that MICP can induce the formation of crystals with exceptional cementing properties through the metabolic activities of microorganisms. When applied to geotechnical materials, this technology significantly boosted their strength [7,8].

MICP technology empowers an ecologically friendly approach to enhance various objects since it can improve the earth and rock stiffness, bearing capacity, permeability, and liquefaction resistance. Song et al. [9] conducted several cycles of MICP grouting on sandstone. They acquired remarkable results: a 229% increase in uniaxial compressive strength, a 179% increase in elastic modulus, and a 177% increase in brittleness index compared to the pre-grouting conditions. The overall mechanical properties and permeability of sandstone were shown to be primarily impacted by the amount of cemented minerals present, which, in turn, directly governs the microscopic distribution of CaCO_3_, amplifying the efficacy of bio-cementation in sandstone. Banik et al. [10] explored the microstructure and mechanical characteristics of standard sand specimens treated with MICP. The resulting stress–strain data and the gain of strength observed under different pore volume cementing fluids provided insight into quantifying the microbial-reinforced sand’s strength and enhancing the low-strain shear modulus. Zhao et al. [11] carried out a solution and cyclic triaxial tests while regulating the concentration of NaCl and observed the gradual decrease of stiffness and cyclic resistance of the additive solids as the NaCl concentration increased. However, these values remained higher than those of unreinforced sand, and it was discovered that the decline in liquefaction resistance resulted from the conversion of calcium carbonate crystals from clusters to single crystals induced by the technology. Xiao et al. [12] leveraged the application of reinforced coral sand with MICP by acclimating microorganisms to various environments to improve their enzymatic activities; a novel three-stage reinforcement method was developed through their study and its ability to increase the strength of MICP-treated objects and the induced calcium carbonate content was confirmed by experimental outcomes. Additionally, suitable cementing solution concentrations were proven to be advantageous in enhancing the reinforcement strength.

MICP technology has shown enormous potential for producing biological bricks [13] that conform to specific strength standards. This technology offers advantages such as low costs, environmental compatibility, and brief reinforcement periods when applied to tailings reinforcement. Unlike traditional cement and chemical grouting, MICP reinforcement grout does not cause intricate concrete tailings to dissociate. Furthermore, the bio-cement produced is simpler to grind, dissociate, and recycle. The effects of bio-cementation and cement treatment were examined and compared by Yin et al. [14] for oil-contaminated sand. The results demonstrate that while the effects of both bio-cementation and cement treatment were diminished under oil pollution, sand treated with MICP was four times stronger than sand treated with cement using the same quantity of cementing fluid. The unconfined compressive strength of sand treated with MICP surpassed 1 MPa after only eight reinforcement cycles, reflecting the tremendous potential of MICP in improving soil and rock in oil-contaminated regions. However, due to the bacterial fluid’s influence on the cementation rate of microbial reinforcement technology, the strengthening effect might be diminished in unexpected environments, resulting in a different curing strength than expected. Therefore, it is necessary to investigate new methods which can regulate MICP to enhance its strengthening effect and extend its engineering applications.

Current research has focused on regulating bacteria to enhance the efficacy of MICP technology. Researchers have explored the control of pH, culture temperature, bacterial types, and concentration. Wen et al. [15] used urease activity monitoring through changes in conductivity and pH values before MICP reaction to maintain consistent MICP application. Results indicated that the conductivity and pH value rapidly increased after mixing bacteria with urea solution. The optimum conductivity range was 1.5–1.8 ms/cm within 60 min, and the pH range was 8.82–9.02. Using a 0.25 m calcium source in reinforced sand samples produced consistent unconfined compressive strength, improving rock and soil engineering performance. In his study on the effects of bacterial concentration and activity on the MICP process under different temperatures, Wang et al. [16] demonstrated that adjusting temperature controlled induced calcium carbonate’s size and crystal form. Urease activity did not decrease at low temperatures, but precipitation amounts were limited. Low temperatures reduced bacterial growth and the precipitation rate of induced calcium carbonate. High temperatures rendered urease inactive, but repeated grouting increased calcium carbonate quantity, optimizing the application field of MICP. Lv, Tang, Zhang, Pan, and Liu [7] explored different calcium sources and magnesium ions’ effects on the treatment of calcareous sand during MICP treatment. Results indicated that calcium acetate had the highest calcium carbonate content, but calcium nitrate or calcium chloride increased solid strength. Solid strength under different calcium sources increased after adding 0.05 M of magnesium. When the magnesium ion concentration reached 0.5 M, using calcium chloride as the calcium source produced the highest strength and calcium carbonate content, providing novel guiding principles for MICP reinforcement technology.

Researchers are exploring various breakthrough points in the regulation process of bacterial induction, including screening new bacterial strains, adding additives, and externalization techniques. Fazelikia et al. [17] analyzed the ecosystem and screened various strains with MICP performance, comparing the induced mineralization functions of each strain through wind tunnel tests and scanning electron microscopes. The tests demonstrated that the diversity of Bacillus strains provides significant potential for environmental adaptation and plays a critical role in resisting soil and rock erosion.

Furthermore, Shan et al. [18] investigated the feasibility of adding activated carbon, thus improving the adsorption rate of bacteria during MICP reinforcement. The results have shown a significant enhancement in the axial strain strength of samples with increased activated carbon content, along with an improvement in bacteria adsorption rate and strength. At 0.75% activated carbon, the liquefaction resistance of sand treated with MICP was significantly improved, which, in turn, helped to enhance the liquefaction resistance of sand strengthened through MICP. Liang et al. [19] explored the strength mechanism of MICP-treated sand by adding different fibers and varying fiber lengths. The results indicate that segment fibers effectively promote the precipitation of calcium carbonate, limit the movement of sand particles, and enhance the strength of the solidified body. However, overly long fibers can lead to uneven calcium carbonate, thus reducing the strength. Zhang et al. [20] conducted experiments measuring the effects of different protein concentrations on MICP-reinforced tailings. The results showed that with the increase in protein concentration, the solidification heterogeneity of tailings increased. When the protein concentration was 5%, the solidification strength of tailings was at its best, with calcite as the primary induced calcium carbonate. Deng et al. [21] improved the effectiveness of MICP-induced precipitation by introducing an electric field, thus increasing the amount of calcium carbonate induced by the microorganisms. The results reveal that the growth and activity of mineralized bacteria were substantially improved under an electric field of 0.5 V/cm, leading to increased calcium carbonate content. Finally, Su et al. [22] have proposed a new technology that improves reinforcement and reduces byproduct emissions in microbial-induced precipitation. By combining MICP with zeolite, the good adsorption function of zeolite is utilized, leading to significant enhancements in reinforcement strength, reduced permeability, and a reduction in ammonia emissions. This innovation represents a significant improvement in standard MICP technology.

The research on regulating bacterial induction has made remarkable progress, particularly in adding organic matrices to effectively modulate calcium carbonate precipitation induced by bacteria. Organic matrices have been found to directly control calcium carbonate’s nucleation, growth, and aggregation. As early as 1984, Wheeler and Sikes [23] discovered that organic matrices, as an initial factor of crystal growth, can regulate crystal formation, provide a lattice template, or stabilize the core surface of calcium carbonate. Paul and Das [24] also observed that the use of organic substrates had a significant effect on biomineralization. Using organic molecules such as curcumin and quercetin as biological templates, researchers successfully synthesized the most unstable vaterite crystal and stable calcite in calcium carbonate, with vaterite being stable in the liquid phase for up to 18 h.

Organic substrates play a critical role in regulating the mineralization process of calcium carbonate in natural organisms. The organic matrix provides a template for the nucleation and growth of calcium carbonate and regulates the crystal orientation, form, and morphology of calcium carbonate crystals [25]. Yang et al. [26] studied the deposition of calcium carbonate in the presence of various organic substrates, such as beta-cyclodextrin glycogen and soluble starch. Calcite crystals, produced without an organic matrix, can be induced by β-cyclodextrin, while coral aragonite crystals are induced as a regulatory template. Similarly, glycogen and soluble starch induce vaterite. The following research discovered that dextran could effectively induce leaf-like aragonite crystals when used as a template [27]. Zhao et al. [28] studied calcium carbonate induced by amylopectin, Mg^2+^, and Fe^3+^. The results showed that amylopectin was instrumental in forming pumpkin-like calcite, while rod-like and dumbbell-like calcite were formed in the presence of Mg^2+^, and calcite with step depression was formed in Fe^3+^. Hosoda and Kato [29] studied the preparation of calcium carbonate under three organic matrix templates: cellulose, chitosan, and chitin.

The results showed that the organic matrix effectively controlled the morphology of calcium carbonate crystals. Zhang et al. [30] studied the regulation of dumbbell-shaped calcium carbonate on bacterial cell templates and proposed the possible particle formation and transformation mechanism. Azulay et al. [31] studied the regulation of ECM protein in extracellular polymer by calcium carbonate and found that the organic matrix affected carbonate nucleation. The calcite nucleus was formed stably in advance under the control of ECM, which aggregated into a calcite crystal form. Azulay and Chai [32] pointed out that biomineralization can significantly change morphology and structure under the guidance of organic molecules and studied the influence of biopolymer additives on the crystal structure of calcium carbonate. Without additives, carbonate crystals mainly exist in the form of calcite, which is the most thermodynamically stable. Kim et al. [33] studied the formation of calcium carbonate by agar and polyacrylic acid at different concentrations. The results showed that the pH could not be controlled at low concentrations, leading to changes in organic matter components forming an anisotropic template, resulting in elliptical calcite forms. In contrast, a slow pH increase at high concentrations would result in spherical calcite. Erceg et al. [34] studied the application of lipids in the formation of calcium carbonate and found that calcium carbonate formed a shell structure under the action of lipids, providing a new idea for adjusting the precipitation process of calcium carbonate.

Previously, researchers focused on utilizing one or more organic matrices as templates for the chemical preparation and control of the crystal form in calcium carbonate crystals. Based on the biomineralization of calcium carbonate induced by microorganisms, the mechanism of researching the addition of organic matrices as regulatory templates in the biomineralization process is relatively complex, and further research is needed. Organic matrices are mainly utilized to adjust calcium carbonate’s crystal type, appearance, and particle size, providing a corresponding means for producing calcium carbonate crystals with unique morphology and ordered structure.

Microbial-induced calcium carbonate precipitation is a complex system, and the specific reaction formulae are shown in Formulas (1)–(5). Bacteria and corresponding ions lead to biominerals that differ from chemically prepared carbonates, exhibiting distinct morphology, structure, thermodynamic stability, and kinetic properties. Furthermore, the reproduction and metabolism of bacteria during induced mineralization generate organic matter, which further influences biomineralization. As a result, biomineralization is affected by numerous factors, and different organic substrates have varying effects on the process.
(1)$$\mathrm{CO}{\left({\mathrm{NH}}_{2}\right)}_{2}+H_{2}O\overset{\mathrm{Urease}}{\to }{\mathrm{NH}}_{2}\mathrm{COOH}+{\mathrm{NH}}_{3}$$
(2)$${\mathrm{NH}}_{2}\mathrm{COOH}+H_{2}O\to H_{2}{\mathrm{CO}}_{3}+{\mathrm{NH}}_{3}$$
(3)$$H_{2}{\mathrm{CO}}_{3}\to {\mathrm{HCO}}_{3}^{2-}+H^{+}$$
(4)$${\mathrm{HCO}}_{3}^{-}+H^{+}+2{\mathrm{OH}}^{-}↔{\mathrm{CO}}_{3}^{2-}+2H_{2}O$$
(5)$$\mathrm{Cell}+{\mathrm{Ca}}^{2+}\to {\mathrm{Cell}-\mathrm{Ca}}^{2+}$$

In order to understand the effect of the organic matrix on the mineralization of calcium carbonate prompted by microorganisms and the influence of microorganism-induced calcium carbonate precipitate on the reinforcement of the tailings, this study focused on studying a functional strain, *Sporosarcina pasteurii*, known for its mineralization capabilities. Initially, the bacteria were cultured using a solution-based experiment. Different concentrations of the organic matrix, sodium citrate (ranging from 0% to 3.5%), were added to the bacterial culture solution to investigate their synergistic effects. The goal was to ascertain the impact of the organic matrix on crystal formation and microstructure during the process of bacterial-induced mineralization, as well as to understand the underlying mechanism by which the organic matrix affects the properties of the sediment throughout this process. The crystal form and microstructure of the induced calcium carbonate samples were characterized using X-ray diffraction, infrared spectrometry, and scanning electron microscopy. After comparing the outcomes of bacteria concentration, calcium carbonate content, and crystal morphology, the optimal dosage of the organic matrix, sodium citrate, was determined. Lastly, employing the bacteria solution with the optimized concentration of organic matrix sodium citrate, tailings reinforcement tests were conducted on tailings and solid samples, incorporating mechanical testing, microscopic detection, and CT scanning. The objective was to explore how the organic matrix reinforces tailings through microbial activity. The results shed light on the role of the organic matrix in regulating the crystal form of calcium carbonate in microbially-induced calcium carbonate precipitation (MICP), facilitating the control of the solidification process in microbial-enhanced tailings and yielding superior reinforcement strength.

## 2. Materials and Methods

### 2.1. MICP Mineralized Solution to Experiment with Sodium Citrate

#### 2.1.1. Purpose

The solution test aimed to determine sodium citrate’s influence on the mineralization reaction, obtain the optimal addition quantity, and provide the essential parameters for the subsequent reinforcement test.

#### 2.1.2. Materials

*Sporosarcina pasteurii*, purchased from the American Type Culture Collection (Rockefeller, MD, USA), was selected as the dominant strain for the mineralization bacteria, numbered ATCC11859.

The liquid medium contains 15 g/L casein peptone, 5 g/L soy peptone, 5 g/L sodium chloride, 20 g/L urea, and 1000 g/L distilled water, while the solid medium was supplemented with 20 g/L AGAR, with a pH of 7.3.

The cementing liquid consisted of urea solution and CaCl_2_ solution, with a volume and concentration ratio of 1:1.

The sodium citrate used in the test was sodium citrate dihydrate intended for industrial use, with the chemical formula C_6_H_5_Na_3_O_7_·2H_2_O.

#### 2.1.3. Methods

The volume fraction of sodium citrate was set at different levels, namely 1.5%, 2%, 2.5%, 3%, and 3.5%, with a control group (pure water). During the experiment, sodium citrate was added directly to the bacterial culture medium and sterilized to enable a collaborative culture experiment with bacteria. The test temperature was adjusted at 30 °C, with an initial pH of 7.3 and an initial medium OD_600_ of 0. The relevant parameters of specific concentration and dosage are shown in Table 1. The experiment lasted for 32 h, with the reaction solution taken every 2 h for determination. The optimal setting parameters were determined by analyzing the concentration of bacteria in the reaction liquid, the amount of calcium carbonate produced, and the crystal morphology, along with other relevant differences.

### 2.2. MICP Reinforced Tailings Experiment

#### 2.2.1. Purpose

An innovative method was introduced to ensure the safety and stability of tailings in the tailings pond by adding organic matrix sodium citrate into the traditional microbial reinforcement process, thereby reinforcing the effectiveness of the tailings even further. Effect tests to explore the practical application and efficacy of organic matrix sodium citrate in microbial tailings reinforcement were conducted by selecting different concentrations of the organic matrix sodium citrate during the solution test stage. The impact of the regulation of sodium citrate on MICP reinforcement of tailings was assessed, and the feasibility of this approach was analyzed based on the reinforcement efficacy observed after applying the regulation.

#### 2.2.2. Materials

The tailings used in the test were sourced from the dry bank of a tailing pond (Hengyang, China) in Hunan Province. The relative density of Gs is 2.499, with a natural dry density ρ = 1.539 g/cm^3^ and a pore ratio of e = 0.78. Tailings were screened for particle size analysis, and the particle size distribution diagram of tailings was drawn, as shown in Figure 1. The grading parameter was calculated as d_10_ = 0.078 mm, the median particle size as d_30_ = 0.162 mm, d_60_ = 0.2 mm, the coefficient of non-uniformity as C_U_ = 2.532, the coefficient of curvature as Cc = 1.661, the coefficient of non-uniformity as C_U_ < 5, the coefficient of curvature as C_C_ > 1, indicating that the tailings particles are more uniform and belong to the sand of poor gradation.

In the reinforcement test, the microorganisms, culture medium, cementing solution, and sodium citrate used are similar to those in Section 2.1.2.

#### 2.2.3. Test Device and Method

A low-cost and convenient test device is designed, as shown in Figure 2. The ambient temperature is controlled at 30 °C, and the initial pH of tailings is 7.0. The grouting concentration and dosage were set as follows: 600 mL of bacteria solution with OD_600_ = 2.0 (containing 3% sodium citrate by volume), 300 mL of calcium chloride (0.5 mol/L), and 300 mL of urea solution (0.5 mol/L). When reinforcing, the bacteria liquid is injected first for infiltration, and then after the bacteria liquid flows out, the cementing liquid is added for reinforcement. The cementing liquid flowing out from the bottom is pumped into the circular reinforcement, which is carried out once a day, and the total reaction time is 14 days.

## 3. Experimental Results of an Aqueous Solution

### 3.1. Changes in Bacterial Concentration

The bacterial concentration variation was reflected by measuring the OD_600_ value of the bacterial solution. The OD_600_ values of *Sporosarcina pasteurii* solution cultured with and without sodium citrate were respectively determined. The bacterial growth curves of each group are shown in Figure 3.

Figure 3 illustrates that under the influence of organic matrix regulation, the five groups of bacteria exhibited an overall effective growth trend under different growth conditions after adding varying amounts of organic matrix sodium citrate. At the early stage of inoculation culture, the inoculated microorganisms began to multiply following a short adaptation to the culture environment. Roughly two hours after continuous culture, the bacterial concentration without adding organic substrate sodium citrate increased the fastest and entered the logarithmic growth stage earlier than the other groups. Conversely, bacterial growth was inhibited in the culture medium with varying concentrations of organic substrate sodium citrate, and the inhibitory effect became more pronounced with higher concentrations. After entering the logarithmic growth phase, the bacterial concentration increased under standard culture conditions. In the organic matrix sodium citrate group, the bacteria concentration increased rapidly at 1.5%, inhibited at 3.5%, and propagated slowly. However, during the middle and late stages of bacterial culture, the growth process of the group after the addition of sodium citrate was prolonged, and the concentration of the added group exceeded that of the normal group in the later stages. After the bacterial solution reached a stable period under standard culture conditions, the concentration of the bacterial solution with lower organic substrate sodium citrate increased rapidly, reaching the maximum value, indicating that sodium citrate can effectively prolong the bacterial culture process and enhance the concentration of the bacterial solution, which came to its maximum under 1.5% addition and again at 3% addition. Nevertheless, the bacterial solution’s concentration increased gradually during the initial phase of the culture, with the propagation speed of bacteria showing an accelerating trend and the highest growth rate observed during the later stages of the experiment. Afterward, the concentration of the bacterial solution reached a maximum value and began to decrease, entering a declining phase.

### 3.2. Changes of pH in a Bacterial Liquid

The pH value of the bacteria solution was measured by pH meter every 2 h. Figure 4 shows the change curve of the pH value of the bacteria solution with time.

Each group’s initial pH was recorded at 7.3, and during the early stages of cultivation, the pH increased linearly to approximately 9.5 around the 6-h mark. During this period, a profusion of bacteria underwent propagation, while the exchange of nutrients facilitated the effective hydrolysis of urea by the bacteria in the cultural milieu. The resulting ammonium ions, in turn, led to a continuous escalation of the pH, consequently rendering the environment increasingly alkaline. A comparison of pH variations among the groups illustrates that the growth rate of the organic matrix sodium citrate bacterial solution group lagged compared to the normal culture group due to the short-term inhibition in the early stage and the process adjustment during the adaptation stage, thus resulting in a deceleration of the maximum pH value. However, after an extended period of cultivation, the pH values of all groups stabilized and were uniform.

In contrast, the pH level of the normal culture group showed a sudden decrease during a later stage, which could likely be attributed to a consequent decline and death of bacteria in the last stage of culture. Overall, adding organic matrix sodium citrate did not cause perceptible harm to the bacterial cells. Instead, it prolonged their reproductive cycle, enabled higher utilization rates, and augmented their capacity to absorb environmental nutrients.

### 3.3. Changes of Urease Activity in Bacterial Fluid

Bacteria-produced urease catalyzes the hydrolysis of urea, thereby increasing the ion concentration in the culture solution and augmenting its conductivity. As such, changes in the solution’s conductivity are utilized to gauge urease activity in the bacterial solution, and the urease activity is thereby measured by the conductivity method.

In order to determine the urease activity of the bacterial solution, 1 mL of the key is mixed with 9 mL of urea solution at a concentration of 1.5 mol/L. Within 5 min of the concoction, the rate of change in the conductivity of the mixture is measured via a conductivity meter. The calculation of the urease activity for the bacterial solution can be ascertained by means of the conversion presented in Formula (6) [21].
(6)U=11.11f

*U* refers to urease activity and *f* refers to the change rate of conductivity.

After determination, the urease activity of each group of bacterial liquid is shown in Figure 5.

From the analytical outcomes portrayed in Figure 5, we can deduce that the overall trend of urease activity variations in each group is congruent with the alterations in OD_600_ values. During the culture process, urease activity exhibited a continuous increment followed by a gradual stabilization. By juxtaposing the organic matrix sodium citrate group with the normal culture blank group, it becomes evident that urease activity in the entire bacterial solution of the former group is higher than that of the latter group. With time, this disparity gradually augmented. The most prolific growth period among all groups lasted approximately 8 h. Still, due to nutrient limitations in the cultural milieu, bacterial proliferation was restricted, which led to a gradual deceleration in the growth rate of urease activity in the bacterial fluid until it had stabilized. In addition, correlated evaluation with OD_600_ alterations revealed a positive association between urease activity and bacterial concentration. Subsequently, the addition of organic matrix sodium citrate elevated bacterial concentration, thereby enhancing the urease activity of the bacterial solution. Upon inspection of the data, we observed that a 3% inclusion quantity led to maximum growth of urease activity in the bacterial fluid without doing apparent harm to bacteria. Furthermore, after adding organic matrix sodium citrate, the duration of hyper urease activity in the bacterial liquid exhibited more stability.

### 3.4. Calcium Carbonate Production

In the cementation test, bacteria liquid was used, and the proportion of urea to calcium chloride was 1:2:2. After these raw materials were mixed and fully reacted, they could be precipitated by standing. The final calcium carbonate precipitate is filtered out. Then, after drying treatment, weighing and measuring are carried out in a row with an electronic balance, and each group’s calcium carbonate production amount is shown in Figure 6.

In accordance with the findings depicted in Figure 6, substantial disparities were observed among the various test groups regarding the induction of calcium carbonate production by the bacteria. Compared to the blank group (no organic matrix was added) under typical culture conditions, wherein the bacteria yielded 25.16 g of calcium carbonate precipitate, the experimental group supplemented with organic matrix sodium citrate showed noteworthy outcomes. The highest calcium carbonate content induced by the bacteria solution, mixed with a cementing solution, was achieved when 3% organic matrix sodium citrate was added. This resulted in an approximately 22.6% increase in calcium carbonate production compared to the blank group. Notably, across all experimental groups, the lowest calcium carbonate content produced by the reaction between sodium citrate and the cementing solution was observed when a 1.5% concentration of organic matrix sodium citrate was added to the bacterial solution. This can be attributed to the lower organic matrix sodium citrate concentration, which promoted bacterial proliferation. However, during the extended culture cycle, the change in the OD_600_ value among each experimental group revealed that the 1.5% organic matrix sodium citrate group entered the decline phase prematurely, resulting in more bacteria deaths than viability. Consequently, the content of calcium carbonate produced after cementation decreased, primarily due to the diminished availability of nucleation sites, thereby significantly reducing the generated calcium carbonate content.

### 3.5. SEM Analysis of Crystals

In order to investigate the impact of sodium citrate on calcium carbonate crystals produced by the Mineral-Induced Crystallization and Precipitation (MICP) reaction, the crystals were subjected to a morphology test via scanning electron microscopy (SEM). The resulting images of the morphology of each group of calcium carbonate crystals, magnified at a scale of 1000 times, are presented in Figure 7.

The scanning electron microscopy (SEM) images of calcium carbonate generated from the ordinary MICP process (Figure 7a) exhibit a range of crystal forms and irregular structures, mainly formed through the cementation of irregular particles and small spherical particles. Nevertheless, after introducing organic matrix sodium citrate, the precipitation morphology of calcium carbonate in each experimental group underwent a marked change. The spherical morphology became dominant, while the interconnection mode became relatively uniform. It could also significantly promote the formation of spherical calcium carbonate at low concentrations of sodium citrate. These results imply the significance of adding organic matrix sodium citrate to microbial-induced calcium carbonate precipitation. The formation of spherical morphology may be attributed to the interaction between the sodium citrate molecules.

Furthermore, spherical particles with relatively small particle sizes can display superior bonding and filling abilities. Specifically, in the experimental group with a sodium citrate content of 3%, the adhesion of large and small particles was relatively robust, surpassing other “point-contact” bonding forms. Thus, adding organic matrix sodium citrate enhances calcium carbonate’s morphology and cementation properties in microbial-induced calcium carbonate precipitation. Because the physical and mechanical properties of calcium carbonate crystal aggregates with a varied microscopic morphology differ significantly, tailings reinforcement experiments controlled by sodium citrate can be further conducted. By optimizing the morphology and structure of crystal products, the physical and mechanical properties of the tailing’s solidified body can be improved. The reinforcement effect of MICP can be enhanced by doing so, which holds practical significance for its application.

## 4. Results of Tailings Reinforcement

### 4.1. Setting of Sodium Citrate Addition Amount in an Organic Matrix

According to the analysis results of the solution test after synergistic culture of organic substrate sodium citrate and bacteria, the most extended growth duration of the bacterial solution and a steady increase in urease activity were observed upon adding 3% organic substrate sodium citrate. Moreover, experimental outcomes also demonstrated that the most significant amount of induced calcium carbonate sediment containing various crystal structures and tight cementation was obtained under the control of 3% organic matrix sodium citrate. Consequently, the appropriate addition mode and amount (3% soluble organic matrix sodium citrate) were determined by the reinforcement test of tailings involving regulation of the calcium carbonate crystal form. A grouting reinforcement test of tailings was subsequently implemented to explore the reinforcement effect of tailings after adding organic matrix sodium citrate. In this test, the mechanical properties of the tailings after grouting, the distribution of calcium carbonate cement types in the tailings, calcium carbonate content, micro-cementation statuses among tailings particles, crystal form and appearance of calcium carbonate crystals, particle size characteristics, and pore distribution of calcium carbonate crystals were analyzed and calculated through shear strength testing and micro-morphology testing (e.g., SEM, XRD, CT scan, etc.) of the reinforced tailings.

By comparing the test results with the joint MICP reinforcement test group, this study explored the mechanism of tailings reinforcement upon the control of organic matrix sodium citrate.

### 4.2. The Triaxial Test Results

The strain-controlled triaxial apparatus based on the Mohr–Coulomb strength theory (as shown in Formula (7)) was utilized for the undrained shear test. Confining pressures of 200 kPa, 250 kPa, and 300 kPa were set, and the axial force was applied immediately when the specimen became stable, resulting in the tailings specimen’s shear failure under undrained conditions. During the test, critical parameters such as the shear strength index of tailings were monitored and analyzed, and their cohesion and internal friction angle were calculated. The stress–strain curves of tailings samples strengthened by each group of tailings under the consolidated undrained triaxial shear test are depicted in Figure 8.
(7)τ=σ⋅tanφ+c

*τ* is shear strength; *σ* is normal stress; *φ* is the angle of internal friction; c is the cohesion.

The stress–strain curve reveals the changing trend of the tailings samples’ stress–strain response under axial load, broadly categorized into four stages. In the initial loading stage, the pores within the tailings sample begin to compact under the axial force, leading to small transverse deformation and a decrease in sample volume over time. This is the pore compaction stage. Moving into the second stage, the load increases and the deviator stress of the sample reaches its peak value, followed by entry into the third stage. In this stage, the deviator stress decreases gradually, and the specimen incurs some destruction. Continuous loading leads to shear failure along cracks within the sample. As the fourth stage begins, the axial deformation increases while the deviator stress decreases and stabilizes gradually. Once the set axial strain value is reached, the specimen undergoes destruction but retains some strength due to the mutual embedding and occlusion between particles.

Compared to the unstrengthened group, the tailings’ peak deviator stress value of ordinary MICP increases strengthened. In contrast, the peak deviator stress value of tailings strengthened with organic matrix sodium citrate has an even more significant change. The Mohr–Coulomb strength theory formula processes triaxial shear strength test data, obtaining the characteristic values of mechanical parameters of tailings samples under different confining pressures, as displayed in Table 2. The cohesive force, internal friction angle, effective stress ratio, and maximum shear stress of the reinforced tailings samples demonstrate a significant increase. Compared to the experimental group without the addition of organic matrix sodium citrate, the cohesive force and shear stress of the tailings experimental group treated with organic matrix sodium citrate show a significant improvement. Therefore, adding organic matrix sodium citrate can effectively enhance the mechanical properties of tailings while improving the effectiveness of MICP reinforcement of tailings.

### 4.3. Comparative Analysis of Micro-Morphology of Cement

To analyze the microstructure of original tailings and reinforced tailings samples, a scanning electron microscope (SEM) was used. As illustrated in Figure 9, the unreinforced undisturbed tailings particles exhibit irregular distribution forms comprising flake, irregular grain, diamond, and other structures. Furthermore, the surface layer displays numerous large pores, representing a granular medium with relatively high permeability. Upon comparing Figure 9b,c, it becomes evident that adding a 3% organic matrix sodium citrate group induces the formation of more tightly cemented calcium carbonate. Compared to the group lacking organic matrix sodium citrate, the calcium carbonate precipitate obtained by adding 3% organic matrix sodium citrate features a diverse range of structures, shapes, and sizes. Furthermore, various calcium carbonate crystals and tailings particles are encapsulated, extruded, and adhered to one another, thus generating irregular reinforced aggregates. This further highlights how organic matrix sodium citrate can effectively enhance tailings’ stability and strength and facilitate tailings’ reinforcement treatment.

Figure 9c shows calcium carbonate completely covers the tailings particles, effectively binding them together. This coverage and cementation generate a specific stacking morphology. Furthermore, the distribution of calcium carbonate crystals is uniform, with no large-area fixed calcium carbonate crystal form. Irregular calcium carbonate crystals interconnect and stack upon each other to improve their tightness, thus enhancing the reinforcement effect. Owing to the complexation of organic matrix sodium citrate, more calcium ions can be adsorbed and accumulated to generate calcium carbonate crystals. This effect reduces the scouring impact of the solution, thus further strengthening cementation. The various calcium carbonate structures attached to the surface of tailings particles during cementation and solidification also combine calcium carbonate and tailings particles.

Moreover, sodium citrate rapidly dissolves in water, forms complexes with calcium ions, and speeds up the formation and precipitation of calcium carbonate. Numerous tests and practices have proved that this approach can effectively enhance the mechanical properties of tailings, reducing the impact of tailings instability on the environment. The reinforcement strategy based on regulating and inducing the formation of calcium carbonate demonstrates excellent feasibility and economy and is significant in reducing tailings pollution while enhancing the ecological environment.

### 4.4. Crystal XRD Detection

Calcium carbonate has three crystal forms: calcite calcium carbonate, aragonite calcium carbonate, and vaterite calcium carbonate. To better understand the structure and morphology of calcium carbonate crystals, an X-ray diffraction technique was utilized to detect the structure and morphology of calcium carbonate crystals. The crystal structure and crystal form distribution of samples can be qualitatively analyzed by comparing the detection results with the known standard chart cards. This study derived the XRD results from the common MICP technique and reinforced tailings samples with 3% organic matrix sodium citrate, as shown in Figure 10. Following analysis, essential discoveries were obtained regarding the crystal structure and crystal form distribution in the samples.

The bacteria solution cultured by common MICP and sodium citrate added with 3% organic matrix was used as a gum in the tailing sand sample. XRD was conducted on the cemented tailings samples, and the data obtained were analyzed. This revealed that when 3% sodium citrate and bacteria solution were added to the tailings samples for reinforcement treatment, the crystal structure of calcium carbonate was mainly calcite calcium carbonate, followed by vaterite calcium carbonate. The dominant crystal planes in the XRD results appeared at (004), (012), (112), (104), (114), (113), (202), (204), and (118), with the (104) crystal plane being the symbolic corresponding crystal plane of calcite calcium carbonate. In contrast, the dominant crystal planes such as (012), (110), (113), (202), and (204) are notable characteristics of calcite calcium carbonate. On the other hand, the corresponding crystal planes of vaterite calcium carbonate are (004), (112), and (114). Therefore, the calcium carbonate cement formed by adding sodium citrate predominantly features calcite calcium carbonate, and its proportion is higher than that of vaterite calcium carbonate mixed crystal. Compared to tailings reinforced by ordinary MICP, adding 3% organic matrix sodium citrate can generate well-formed calcium carbonate precipitated crystals. The proportion of calcite and vaterite is more reasonable, and the mixed crystals can have more effective cementation within tailings.

### 4.5. Crystal FT-IR Detection

Two techniques were utilized to investigate the reinforcement effect of tailings: typical microbial-induced calcium carbonate precipitation (MICP) reinforcement and reinforcement by adding 3% organic matrix sodium citrate. Samples of reinforced tailings were analyzed using infrared absorption spectroscopy. The corresponding crystal form structure of calcium carbonate was successfully obtained through the detection and analysis of the wave values under the infrared characteristic absorption peaks of each crystal form of calcium carbonate in the sample and by comparing the resulting spectrum with the standard spectrum of wave numbers.

The Figure 11 infrared spectra demonstrate that the blank group exhibits distinctive absorption peaks at 3419.79 cm^−1^, 2509.39 cm^−1^, 1406.83 cm^−1^, 1076.28 cm^−1^, 878.68 cm^−1^, and 711.73 cm^−1^ [35,36]. The absorption peaks at 2509.39 cm^−1^ and 3419.79 cm^−1^ are primarily due to the O-H bond’s symmetrical and asymmetrical stretching vibrations, resulting from the hydroxyl groups and adsorbed water of calcium carbonate particles. Conversely, V_4_ and V_2_ characteristic absorption peaks of calcite crystals at 711.73 cm^−1^ and 878.68 cm^−1^ correspond to V_4_ of vaterite crystals at 1076.28 cm^−1^, respectively. Notably, the IR spectra of calcium carbonate reinforced with 3% sodium citrate register peaks at 1416.03 cm^−1^, 1083.03 cm^−1^, 873.57 cm^−1^, and 711.85 cm^−1^. Upon comparison to standard spectra, gravimetric analysis reveals the calcium carbonate reinforced with 3% sodium citrate to be composed of calcite and vaterite. The corresponding wave numbers of calcite are 1416.03 cm^−1^, 1083.03 cm^−1^, and 711.85 cm^−1^; the corresponding peaks of calcite are relatively protracted, signifying that the proportion of calcite calcium carbonate in calcium carbonate exceeds that of vaterite calcium carbonate. This outcome coincides with calcium carbonate crystal’s X-ray Diffraction (XRD) findings. The addition of 3% organic matrix sodium citrate can regulate the calcium carbonate crystal to reinforce the tailings, and consequently, the crystal form of calcium carbonate can be adjusted.

### 4.6. Quantitative Characterization of Pore Structure Based on CT Scanning

The tailings samples reinforced by typical microbial-induced calcium carbonate precipitation (MICP) and microbial grouting with 3% organic matrix sodium citrate were scanned layer by layer using CT technology. The result data were reconstructed using three-dimensional visualization software ( Dragonfly 2021.1 Build 977), and digital image information was established. By utilizing this microscopic research method, we explored the microscopic changes in the pore structure of tailings following grouting.

To reconstruct the tailings specimen and obtain a quantitative characterization of its pore structure, the data generated by layered scanning were imported into Dragonfly software for further analysis. The Dragonfly software was employed to effectively remove any artifacts and execute multiple iterative reconstruction algorithms to achieve the stereo reconstruction of the data. The resulting three-dimensional reconstruction of the reinforced tailings specimen subjected to ordinary MICP and 3% organic matrix sodium citrate is depicted in Figure 12a,b. CT technology provided a powerful means to reveal the characteristics of the microstructure change of the tailings following grouting reinforcement. This technique yields significant theoretical and technical support for the subsequent optimization design and facilitates a comprehensive understanding of the changes in pore structure following tailings reinforcement.

The distribution of pores and tailings particles in the tailings samples after grouting reinforcement can be distinguished through the grey values generated by CT scanning in two-dimensional slices of the three-dimensional sectional image. The pore structure of the tailings samples can be obtained by segmented threshold processing of the cut data volume through Dragonfly software and binary conversion, as illustrated in Figure 12b,c. By analyzing the three-dimensional pore diagram, it can be inferred that the tailings strengthened with 3% organic matrix sodium citrate have significantly fewer pores than those strengthened through ordinary MICP. In contrast, the upper part of the tailings strengthened via ordinary MICP has more pores than the lower part, which may be a result of repeated grouting and slurry erosion-generated deposition and cementation of calcium carbonate particles, leading to a non-uniformity of the tailings sample during ordinary MICP reinforcement. Compared to ordinary MICP, the overall porosity of the tailings reinforced via adding 3% organic matrix sodium citrate is considerably lower, especially in the upper and middle parts of the reinforced samples. The integrity of the samples becomes robust, and the strengthening effect is also significantly improved. This improved behavior may be attributed to the role of organic matrix sodium citrate in enhancing the activity and proliferation of bacterial liquid and providing more nucleation sites for calcium carbonate crystal-induced precipitation during cementation. The pores in the middle and upper parts of the strengthened samples are lower than those in other parts. However, the related mechanism of organic matrix regulation in MICP requires further exploration and improvement.

The Dragonfly software is capable of analyzing the two-dimensional slice diagram of tailings samples in different layers along the Z axis (from the bottom to the top of the tailings samples), layer-by-layer, to obtain the statistical results of porosity of cross-sections, as illustrated in Figure 13.

Based on the analysis of the results depicted in Figure 13, it can be observed that the porosity of the tailings column samples exhibits an overall upward trend, with the porosity gradually increasing from the top to the bottom. In the standard MICP reinforcement test without ammonium citrate addition, the porosity of the tailings column fluctuates between 6.85% and 28.21%, with an average value of 12.93%. However, in the reinforcement test using 3% organic matrix sodium citrate, the porosity of the tailings column ranges only from 2.15% to 11.25%, with an average value of 6.54%. Compared to the standard reinforcement test group, the porosity of the tailings reinforced by grouting under organic matrix ammonium sodium citrate control is relatively low, reduced by 49.42%. This indicates that sodium citrate can effectively expedite the precipitation process of calcium carbonate induced by microorganisms.

Additionally, because the reinforcement slurry is injected from the top of the sample, the infiltration may remove some fine tailings and calcium carbonate sediments, ultimately increasing surface pores. Generally speaking, adding an organic matrix can effectively decrease the porosity of tailings samples, particularly in the middle and upper parts. The strengthening velocity is significantly faster than that of ordinary MICP, and the cementation uniformity is enhanced.

## 5. Discussion

The organic matrix plays crucial roles in regulating microbial-induced calcium carbonate precipitation (MICP) and in promoting the reinforcement of soil and rocks with the aid of microorganisms. By providing sufficient nutrients and energy, the organic matrix stimulates the growth and metabolism of microorganisms. It promotes the precipitation of calcium carbonate, ultimately leading to the formation of robust cementation and an increase in the strength and stability of soil and rock. Through its hydrolysis and biodegradability, the organic matrix modifies the physical and chemical properties of soil and rock, improving the living environment of microorganisms and enhancing their adaptability and stress resistance, thus sustaining their performances in diverse environments. In addition, the organic matrix has various sources and types that can be proportioned and modified to suit different rock and soil types and reinforcement requirements, allowing for high flexibility.

Adding an organic matrix, specifically sodium citrate, to the experimental setup had a notable positive impact on the efficacy and quality of the MICP reinforcement. Comparative analysis revealed that introducing an organic matrix in the sample could lead to a reduction in porosity and an enhancement in the strength and stability of the tailings specimens. Nawarathna et al. [37] also employed an organic matrix, namely chitosan, to augment the amount of induced calcium carbonate precipitation and subsequently solidify sand—the resultant precipitation comprised calcium carbonate and chitosan hydrogel. Implementing the organic matrix, chitosan, altered the morphology of calcium carbonate crystals, yielding a more robust and consolidated structure than traditional methodologies. Nawarathna et al. [38] investigated the regulation aspects of biomineralization using two different organic substrates. The graphical representation of the relationship between the induced precipitate amount and the poly (L-lysine) concentration exhibited a bell-shaped curve, indicating a concentration-induced regulatory and inhibitory relationship. Conversely, variations in the polyglutamic acid concentration had no impact on the precipitate quantity. Under poly L-lysine’s influence, calcium carbonate crystals’ morphology shifted from well-developed rhombohedral crystals to elliptical aggregates.

In contrast, when polyglutamic acid is introduced, polyhedral and spherical crystals will predominate. Despite a small quantity of organic matrix being added, its presence exerts control over nucleation and effectively manages the size, shape, structure, and growth orientation of calcium carbonate crystals. Additionally, it governs the nucleation site responsible for forming specific calcium carbonate crystal forms [39]. However, certain concerns are associated with utilizing organic matrix sodium citrate. Firstly, controlling the amount of sodium citrate within a specific range during induced precipitation is crucial. Excessive amounts of sodium citrate can hinder microbial growth and diminish the desired reinforcement effect. Secondly, due to the biological nature of the organic matrix, attention must be given to the potential issue of bacterial contamination during its application to prevent adverse impacts on the environment.

The organic matrix assumes a significant role in the microbial fortification of rock and soil. Still, meticulous attention ought to be directed toward regulating the supplementary quantity and bacterial contamination during its implementation. The successful application of MICP reinforcement necessitates the perpetuation and efficacy of calcium carbonate precipitation to be upheld. The outgassing of ammonia and subsequent biological oxidation of ammonia, serving as the principal byproducts of urea decomposition, will diminish the concentration of ammonium in the surroundings, consequently resulting in a decline in pH value, which may induce a marginal dissolution of the induced calcium carbonate [40,41]. In forthcoming investigations, further discernment of the interconnectedness between the organic matrix, microorganisms, and cement strength must be explored to amplify the utilization of MICP technology in engineering practice.

Moreover, novel formulations and production techniques for organic matrices can be delved into, thereby forging more environmentally sustainable, efficacious, and cost-effective microbial reinforcement technologies for rock and soil. Additionally, the bedrock of research pertaining to the microbial bolstering of rock and soil can be fortified, encompassing the exploration of bioactive substances and a persisting in-depth understanding of the consolidation mechanism and regulatory pattern of microorganisms. Enhancing the reinforcement efficacy and widening the scope of application can expedite the extensive implementation of geotechnical engineering.

## 6. Conclusions

To enhance the efficacy of microbial reinforcement of tailings, the addition of organic matrix sodium citrate is implemented in the experiment to bolster the solidification of tailings through microbial-induced calcium carbonate precipitation (MICP). The research set out to determine the optimal amount of organic matrix sodium citrate and the bacterium regulation effects of sodium citrate via a combination of solution and solidification tests. Through an analysis of the corresponding solidified tailings samples’ shear strength values and microscopic inspection results, investigators were able to evaluate the effectiveness of tailings reinforcement through organic matrix regulation and reach the following conclusions:(1)During the solution test, it was discovered that adding organic matrix sodium citrate positively impacted the early stages of bacterial proliferation, which extended the culture period of the bacteria solution. The bacteria solution concentration improved somewhat in the later stages compared to the control group. The most significant growth of bacteria was observed when 3% organic matrix sodium citrate was added.(2)The change in urease activity and pH value of the bacteria solution was consistent with the change in OD concentration, and the addition of 3% organic matrix sodium citrate resulted in optimal and stable urease activity of the bacteria solution. Furthermore, based on the analysis of calcium carbonate sediment crystal structures induced by different amounts of organic matrix sodium citrate, adding 3% organic matrix sodium citrate resulted in the highest calcium carbonate content and a greater vaterite prevalence. In contrast, the cementation products in the control group exhibited smaller particle sizes, and no morphology was fixed. The sediments in the organic matrix group displayed various crystal forms and tight cementation. Based on these outcomes, 3% organic matrix sodium citrate was deemed the optimal addition amount for subsequent testing parameters.(3)In the curing test, tailings treated with 3% organic matrix sodium citrate displayed significantly enhanced peak stress and effective stress ratio after reinforcement, corresponding with increases in cohesion and internal friction angle values to 153.298 kPa and 42.364, respectively. These findings indicate that adding 3% organic matrix sodium citrate can effectively improve the mechanical properties of microbial-reinforced tailings.(4)In the SEM microscopic test, the distribution of calcium carbonate produced under the influence of organic matrix sodium citrate was uniform in the tailings samples. Additionally, the tailings particles were entirely covered and bonded in a particular stacking form, significantly improving the sample’s compactness. XRD and IR data revealed that the crystal structure of calcium carbonate cement produced by 3% sodium citrate included both calcite and vaterite types, with calcite type calcium carbonate dominating the structure, and the polycrystalline composite structure strengthened the tailings’ reinforcement effect. The CT scanning test demonstrated that the porosity between the tailings was effectively reduced under the influence of organic matrix sodium citrate, with a 68.49% reduction in porosity compared to ordinary microbial reinforcement. This resulted in a significant decrease in delicate pores, improving the cementing effect and demonstrating the uniformity of the reinforced tailings.

## Figures and Tables

**Figure 1 materials-16-05337-f001:**
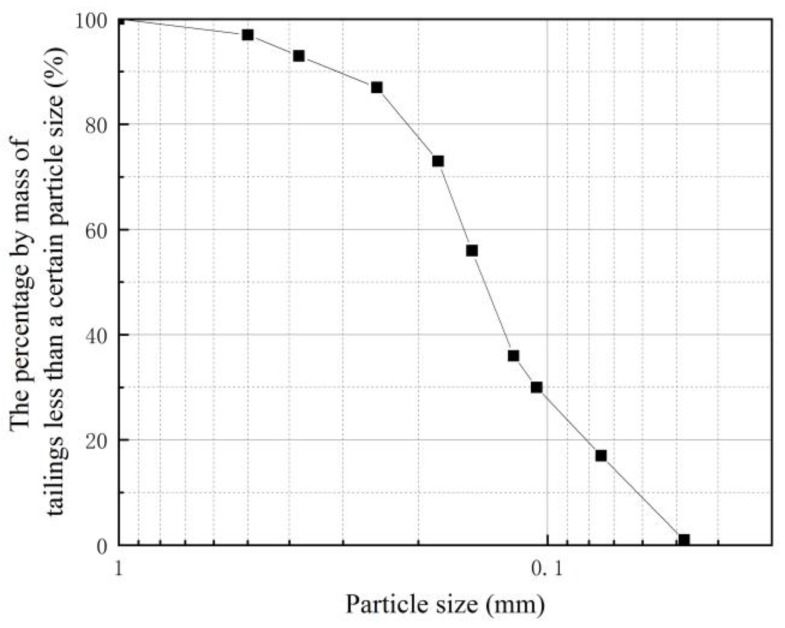
Grading curve of tailings.

**Figure 2 materials-16-05337-f002:**
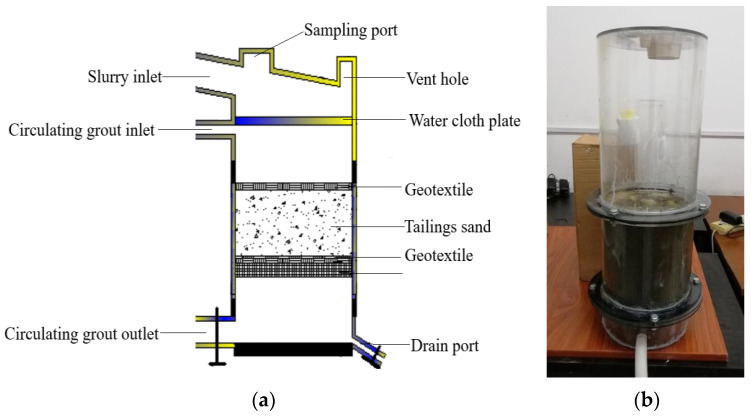
Schematic diagram of experimental apparatus. (**a**) Schematic diagram; (**b**) Experimental apparatus.

**Figure 3 materials-16-05337-f003:**
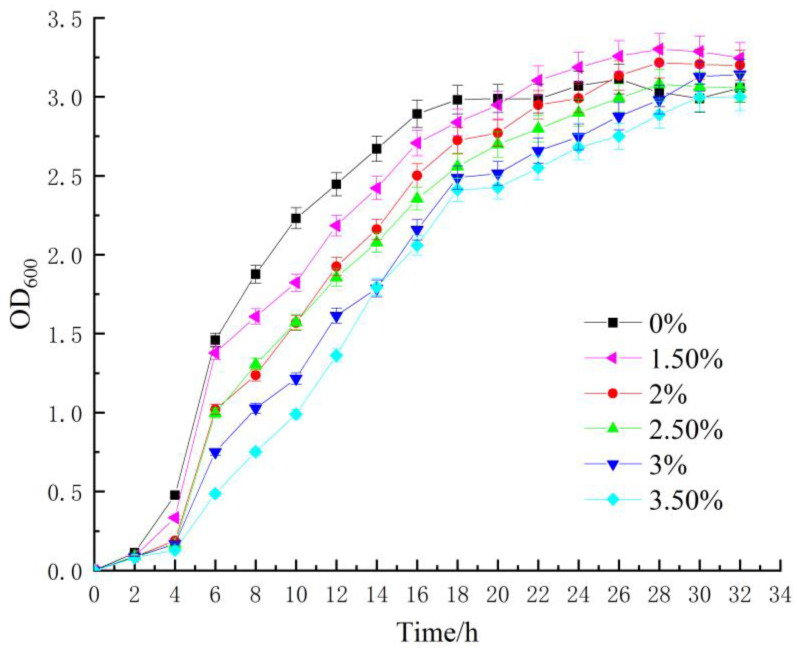
Changes of bacterial concentration under the action of sodium citrate in different organic substrates.

**Figure 4 materials-16-05337-f004:**
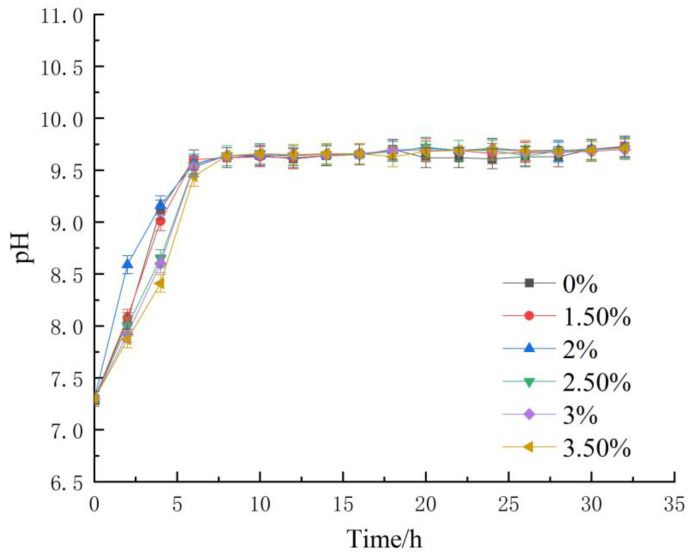
Changes in pH.

**Figure 5 materials-16-05337-f005:**
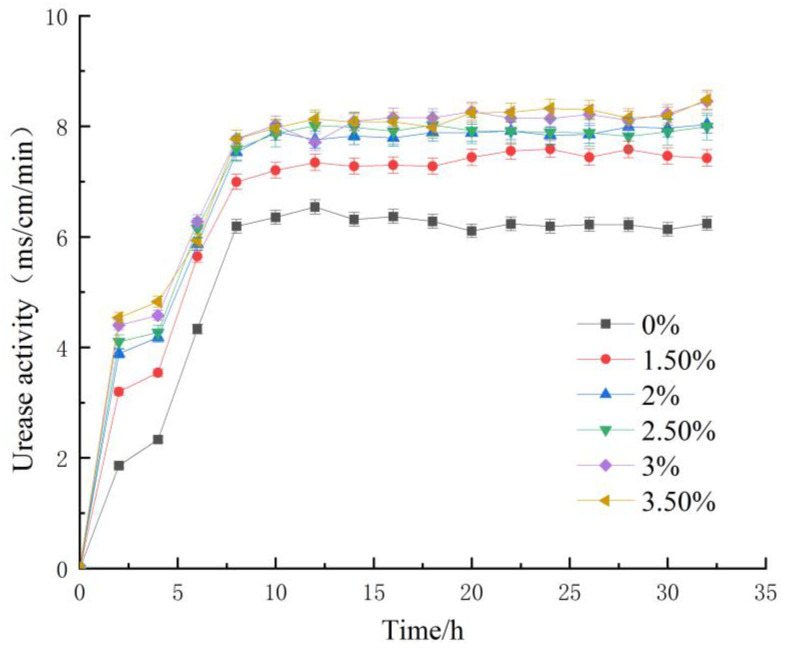
Change of urease activity with time.

**Figure 6 materials-16-05337-f006:**
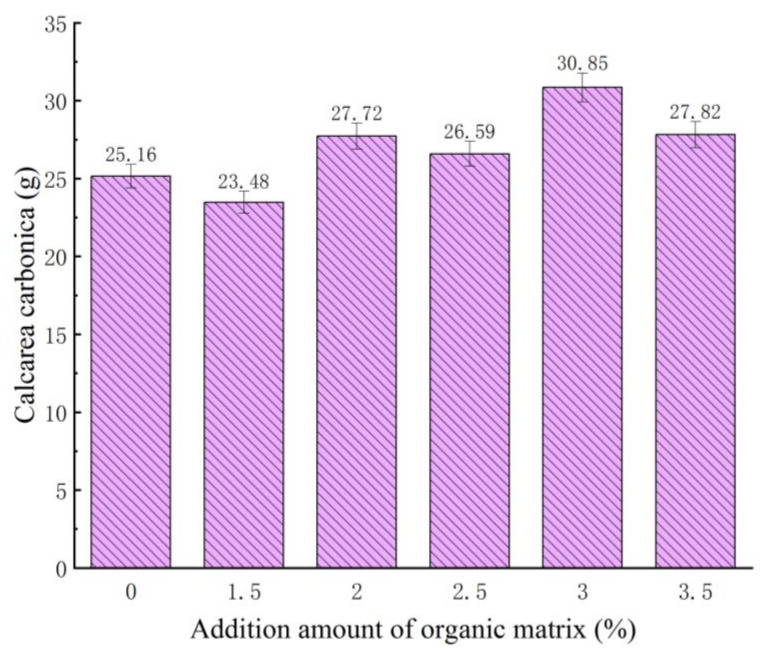
Calcium carbonate production in each group.

**Figure 7 materials-16-05337-f007:**
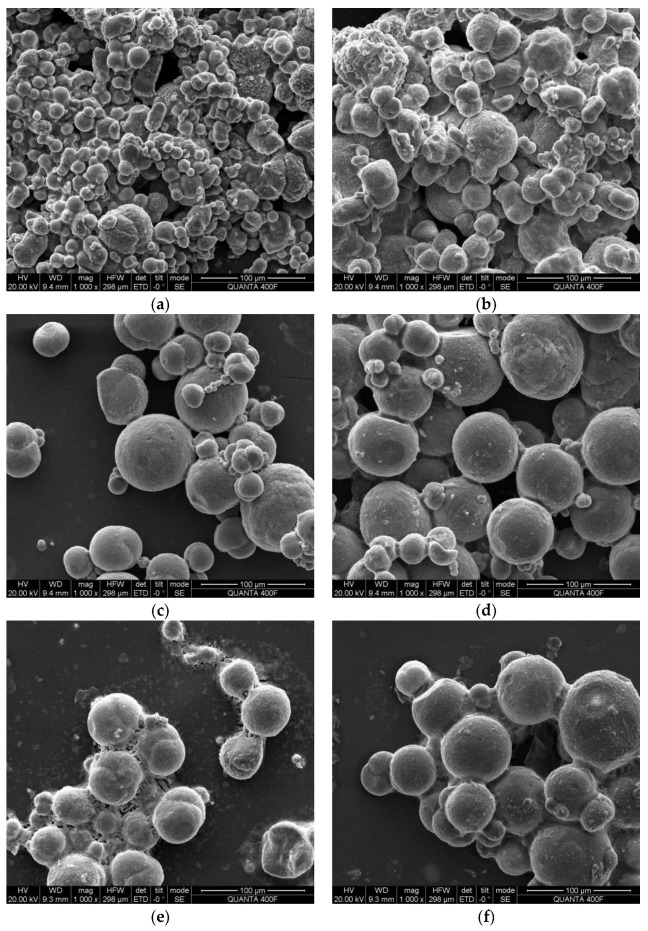
SEM morphologies of each group of calcium carbonate crystals were magnified 1000 times. (**a**) 0%; (**b**) 1.5%; (**c**) 2%; (**d**) 2.5%; (**e**) 3%; (**f**) 3.5%.

**Figure 8 materials-16-05337-f008:**
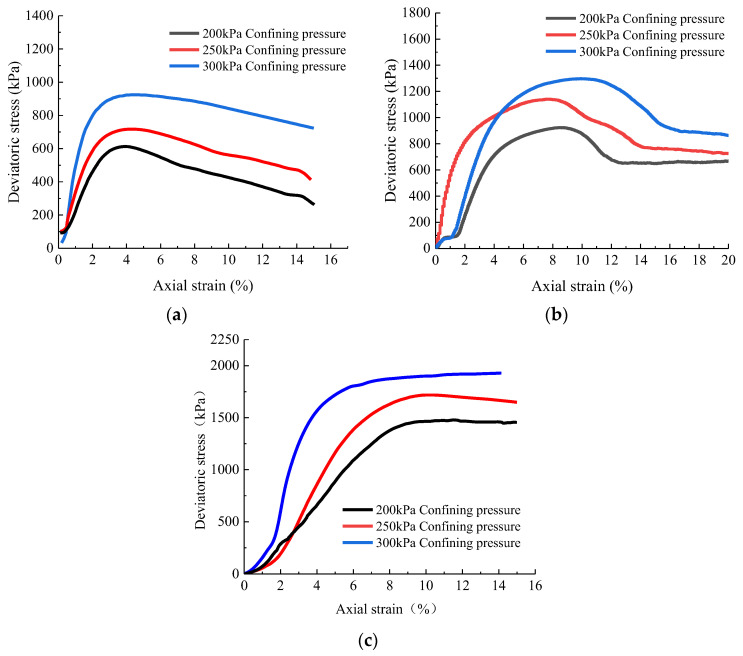
Stress–strain relation curve of tailing samples. (**a**) Undisturbed tailings. (**b**) Ordinary MICP reinforcement. (**c**) Tailing specimens strengthened by MICP with 3% organic matrix sodium citrate.

**Figure 9 materials-16-05337-f009:**
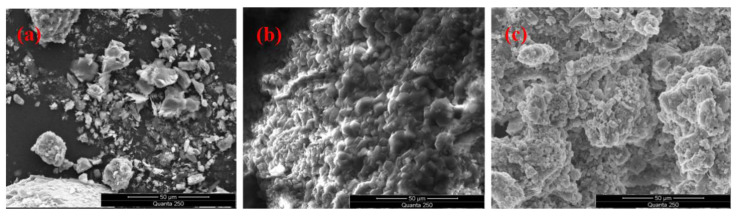
SEM images of the tailings before and after reinforcement. (**a**) Unconsolidated tailings. (**b**) MICP-reinforced tailings without organic matrix sodium citrate. (**c**) Strengthening tailings with 3% organic matrix citric acid.

**Figure 10 materials-16-05337-f010:**
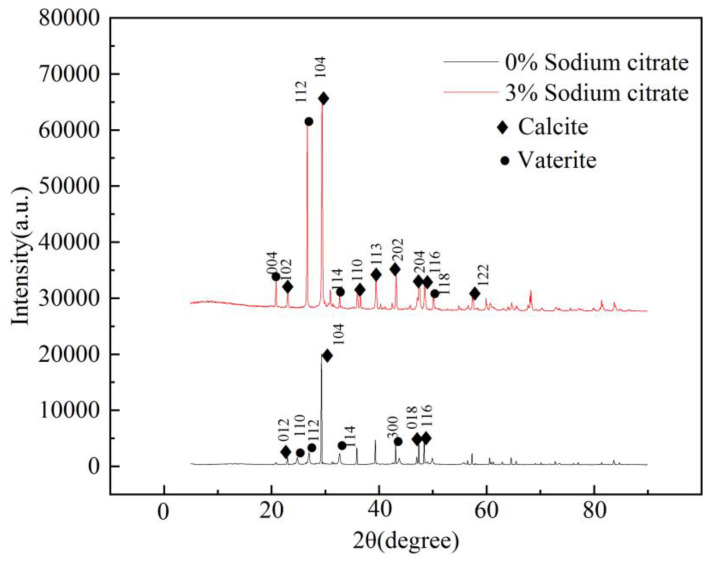
XRD detection diagram of tailings (peak of calcium carbonate).

**Figure 11 materials-16-05337-f011:**
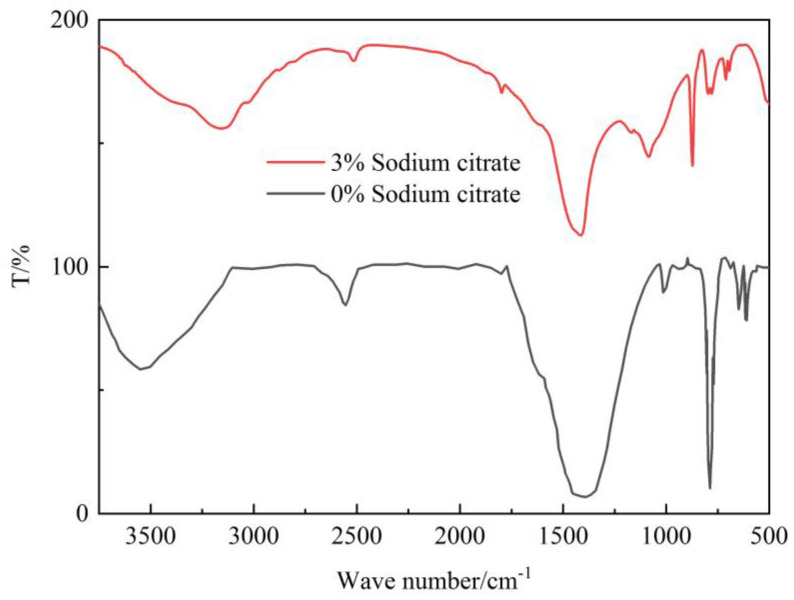
FT-IR detection diagram.

**Figure 12 materials-16-05337-f012:**
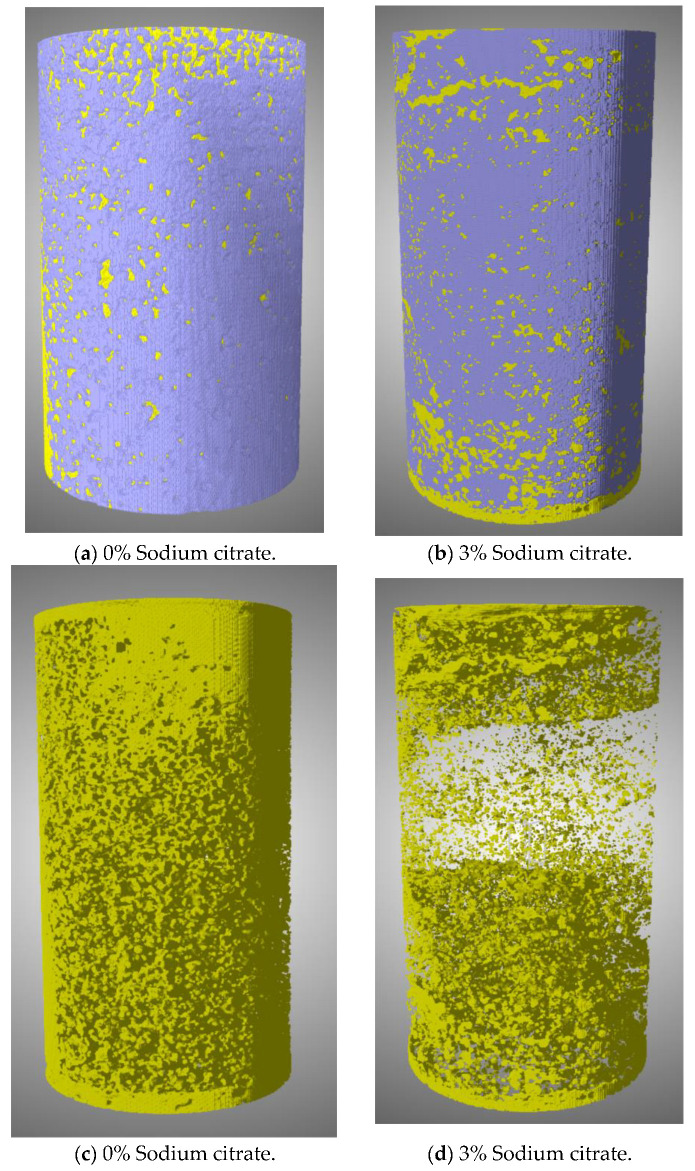
Three-dimensional diagram and three-dimensional pore structure of tailing samples. (**a**) Three-dimensional reconstruction of tailings samples strengthened by common MICP (0% sodium citrate). (**b**) Three-dimensional reconstruction of tailings samples strengthened by MICP with 3% organic matrix sodium citrate group. (**c**) Three-dimensional porosity reinforcement of tailings samples by common MICP (0% sodium citrate). (**d**) Three-dimensional porosity reinforcement of tailings samples by MICP with 3% organic matrix sodium citrate group. Description: Yellow represents pores; Purple stands for tailings.

**Figure 13 materials-16-05337-f013:**
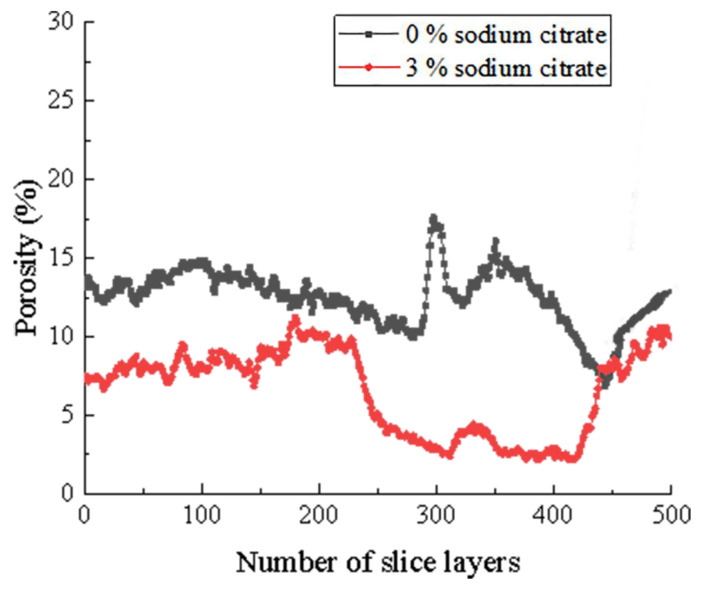
Statistics of surface porosity of tailings sample layer-by-layer.

**Table 1 materials-16-05337-t001:** Parameters of each material.

Solution Test * 1	Parameters
*Sporosarcina pasteurii* bacterial liquid	100 mL
Usage of cementing fluid	400 mL
Sodium citrate volume ratio	1.5–3.5%
Urea concentration	0.5 M
Solution concentration of CaCl_2_	0.5 M

* represents a single trial group.

**Table 2 materials-16-05337-t002:** The strength and stress characteristic parameter table of tailings.

Experimental Group	Confining Pressure (/kPa)	Deviatoric Stress (/kPa)	Cohesion (/kPa)	Internal Friction Angle (/°)	Effective Stress Ratio	Shear Stress (/kPa)
Primary tailings	200	416	4.785	41.081	3.08	308
	250	468			2.874	359.3
	300	620			3.064	461.4
MICP 0% Sodium citrate	200	923	41.739	40.679	5.62	561.5
	250	1140			5.56	695
	300	1298			5.33	799
MICP 3% Sodium citrate	200	1317	153.298	42.364	7.57	758.6
	250	1484			6.93	867.4
	300	1630			6.42	965.2

## Data Availability

The data presented in this study are available on request from the corresponding author.

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
