# Peer review of "Experimental Study on the Effect of an Organic Matrix on Improving the Strength of Tailings Strengthened by MICP"

_materials, 2023, doi:10.3390/ma16155337_

Round 1
Reviewer 1 Report
- The abstract need to be rewritten.
- The introduction is very shallow and not even up-to date.
- What is the gap of the study?
- what are the codes of practice that you followed?
- Check line 107.
- What is the function of section 2.2.1?
- More results are required.
- The discussion is not critical and academically lacking.
- There are many typos and grammatical errors.
- Sections 4.1 to 4.5 are very superficial.
- Conclusion is very simple and general.
- The abstract need to be rewritten.
- The introduction is very shallow and not even up-to date.
- What is the gap of the study?
- what are the codes of practice that you followed?
- Check line 107.
- What is the function of section 2.2.1?
- More results are required.
- The discussion is not critical and academically lacking.
- There are many typos and grammatical errors.
- Sections 4.1 to 4.5 are very superficial.
- Conclusion is very simple and general.
Reviewer 2 Report
This is an interesting paper using bacteria to produce calcium carbonate particles in the presence of citrate. The results are interesting and the paper should be published after some revision, as follows:
- The amount of CaCO3 produced under different amounts of citrate (Figure 4) does not follow any defined trend – it is clearly not dependent on the concentration of citrate present. It looks like citrate does not really affect the amount of mineral produced, can the authors comment on this? Are the differences between each concentration just due to biological variability? How many samples were measured? The bars need to have error bars to show if the differences are significant or not.
- The SEM images in figure 5 are too small, it is not possible to see even the scale bar. This is important given that the authors claim that citrate increases the size of the particles. The images need to be made larger and the scale bars clearly visible, otherwise it is not possible do compare the particle sizes.
- It is not clear how the proportion of calcite and aragonite in each sample was determined using XRD. The authors need to explain this, and provide another technique showing that. As far as the XRD pattern shows, one can only see that there is a mixture of these two phases.
- The authors are assuming that the spherical particles are aragonite, but one cannot be sure. The SEM images in this manuscript shows that all CaCO3 particles are spherical, regardless of the presence of citrate. Is this an effect of the conditions of CaCO3 formation? The authors should do Raman spectroscopy on individual particles to determine the polymorph type.
Reviewer 3 Report
The idea of this work is so interesting, and the authors have done a series of experiment to answer their objectives. However, the explanation and discussion of the results are still insufficient to convince the readers in general. I would recommend substantial revisions as follows:
1.) Why did the bacterial concentration start at OD = 2.0?
2.) Typo bacterial name in Table 1
3.) Delete “Figure 1” below Table 1
4.) Please recheck how to appropriately write scientific name of the bacteria throughout the manuscript.
5.) Rename Figure 3 caption. It is the concentration of sodium citrate that was changed, not the bacterial solution concentration.
6.) Method 2.1.3 indicated that the starting OD of bacterial culture was 2.0, but the starting OD of bacterial growth in Figure 3 is 0 at Time 0. Why is that so?
7.) Considering that the bacterial growth was observed within urea medium, in which the precipitation of CaCO3 was induced, the precipitated CaCO3 should have interfered with the OD measurement. The trend of bacterial growth in Figure 3 may not directly reflect the growth of bacteria alone, possibly leading to misinterpretation of the result.
8.) By considering Figure 3&4, this sentence “The results indicated that the addition of sodium citrate did not inhibit the bacteria activity, and the mineralization proceeded normally.” is not true. Both the growth and mineralization changed with changing concentration of sodium citrate. Please re-explain.
9.) In most cases, S. pasteurii produced CaCO3 in the form of rhombohedral crystals. Why, in this work, did the bacteria produce spherical CaCO3? Did XRD was carried out for the samples in Figure 5? Please discuss.
10.) Why was only 3% sodium citrate selected for further experiments (e.g., triaxial test)? What about the other conc. Of sodium citrate? Please discuss.
11.) The number indicating X-axis in Figure 8 must be wrong. The highest peak of calcite appears at 30ÌŠ 2theta! Please recheck.
12.) It is still doubtful that calcite and aragonite in Figure 8 are really related to the crystals in Figure 5 or not?
13.) Figure 11, Why did %porosity suddenly increase in 0%sodium citrate sample after 450 slices? Please discuss.
14.) As more discussion in each section is highly recommended, more references should be added.
Reviewer 4 Report
Overall, I think this is interesting research with potentially interesting results for the scientific community. However, I think the document looks more like a technical report than a scientific paper. What struck me most is the lack of discussion of the results obtained.
Introduction
The introduction should address why choosing a sodium citrate matrix is a good alternative. If it is the first time it is used in this application it could also be explicitly stated.
I have a doubt, can the bacteria use this organic matrix for growth?
Materials and methods
The information regarding the experimental design and materials used is clear, however, the analytical methodology is not clearly indicated. For example, about the DO, how was it measured? what equipment was used?.
With respect to the cementitious liquid, what concentrations of urea and calcium chloride were used?
How was the mass of precipitated calcium carbonate determined?
How was shear strength performed?
Urea consumption measurements were performed?
Results
3.1: Microbial growth rates during the exponential phase could be calculated. It is basic and simple, but allows quantitative comparison of the effect of the sodium citrate matrix.
3.2: Is the mass of calcium carbonate precipitated as expected? If mass balances are done, how much of the initially added calcium precipitates?
3.3: if crystals are indicated to be larger or smaller, quantitative measurements should be made for comparison.
No discussion of results!
Some specific comments follow:
Line 84: Bacterial names must be in italics.
Line 107: The word figure appears without any associated image.
Line 137 and Table 1: Bacillus pasteuris is an old name for S. pasteurii.
Figure 3: more details should be provided in the figure legend, e.g. what the colors mean.
Line 164: To accurately indicate that there are significant differences between results, statistical analyses such as ANOVA must be performed.
Line 189-191: That information should be in the figure legend.
Figure 6: The y-axis dimension should be the same for all 3 charts. This would facilitate the comparative analysis between them.
Line 197: Where is the formula? If it is important for interpreting the results, it should be indicated.
Line 262: What is CT technology, how it is performed: Materials and methods.
Round 2
Reviewer 1 Report
Accept
Author Response
Thank you for your careful review of the manuscript; your suggestions have greatly improved the quality of the manuscript; I look forward to the follow-up research results and can also get your guidance; thank you again.
Reviewer 2 Report
All my concerns were addressed.
Some minor revisions on the language are appropriate.
Author Response
Thank you very much for your professional guidance, which makes the paper more scientific and readable. Some changes have also been made to the language of the manuscript. Once again, I would like to express my heartfelt thanks for your help.
Reviewer 3 Report
1.) Details in topic 2.1.1 should be presented in the Introduction.
2.) Revise English language throughout the manuscript is highly recommended.
3.) In the revised manuscript, Figure 3 showed that the bacteria grew best at 0% sodium citrate, but Figure 5 showed that the highest urease activity was found in 3.5% sodium citrate culture. Confusingly, the authors explained that the result in Figure 5 correlates with that of in Figure 3! Please revise the discussion and explain why these results are not consistent.
4.) Figure 6 (in the revised manuscript) clearly showed that there were a significant amount of CaCO3 produced in the urea medium (0% sodium citrate), but the authors replied to the comment saying "Considering that the bacterial growth was observed within urea medium, in which the precipitation of CaCO3 was induced, the precipitated CaCO3 should have interfered with the OD measurement. The trend of bacterial growth in Figure 3 may not directly reflect the growth of bacteria alone, possibly leading to misinterpretation of the result." that there was no CaCO3 produced in the urea medium, so the authors selected to use OD value as an indicator for bacterial growth. Please clarify this point carefully!
5.) Please recheck X and Y axis names and units of Figure 6.
6.) Parts of the Discussion appear in the Result part but without any reference! (e.g., references for FT-IR interpretation must be cited) Please cite other literature as appropriate to make the elaboration more convincing.
Revise English language throughout the manuscript by native speaker is highly recommended. There are also some scientific words that were misused (e.g., co-culture).
Reviewer 4 Report
The document is much improved, but the introduction is too long and some of the information provided could be used to directly support the discussion section.
Author Response
Thank you very much for your help; your guidance for the improvement of the quality of the manuscript is very significant; the relevant changes are uploaded in the form of an attachment, again, thank you very much for your help.
